# Volcanic crystals as time capsules of eruption history

Teresa Ubide [1,2] & Balz S. Kamber[1]

Crystals formed prior to a volcanic event can provide evidence of processes leading to and timing of eruptions. Clinopyroxene is common in basaltic to intermediate volcanoes, however, its ability as a recorder of pre-eruptive histories has remained comparatively underexplored. Here we show that novel high-resolution trace element images of clinopyroxene track eruption triggers and timescales at Mount Etna (Sicily, Italy). Chromium (Cr) distribution in clinopyroxene from 1974 to 2014 eruptions reveals punctuated episodes of intrusion of primitive magma at depth. Magma mixing efficiently triggered volcanism (success rate up to 90%), within only 2 weeks of arrival of mafic intrusions. Clinopyroxene zonations distinguish between injections of mafic magma and regular recharges with more evolved magma, which often fail to tip the system to erupt. High Cr zonations can therefore be used to reconstruct past eruptions and inform responses to geophysical signals of volcano unrest, potentially offering an additional approach to volcano hazard monitoring.

[1] Department of Geology, Trinity College Dublin, Dublin 2, Ireland. [2] School of Earth and Environmental Sciences, The University of Queensland, Brisbane QLD 4072, Australia. Correspondence and requests for materials should be addressed to T.U. (email: t.ubide@uq.edu.au)

Arrival of new magma into an existing reservoir is widely acknowledged as a common eruption trigger[1–5]. Mixing of evolving crystal-melt mushes with primitive, hot, undegassed magma may overcome the physical barriers of melt mobilisation and result in a volcanic eruption at the surface[6]. However, the mechanisms and efficiency of blending of injected and resident magmas in the reservoir depend on local physical parameters, controlling degrees of melt hybridisation and crystal recycling[2,3,7] and ultimately variable success rates in eruption triggering[8,9]. Eruption triggers can be monitored geophysically, but the timescales from magma recharge to eruption can be very short, on the order of days to a few years[6,10–12], and the detailed physical pathways from mush rejuvenation to eruption remain largely unconstrained[12]. Accordingly, a better understanding of the recharge mechanisms, and the pathways and timescales on which these occur are needed to improve eruption forecasting at individual volcanoes.

Mt. Etna is the highest (3340 m) and most active volcano in Europe. It has experienced renewed eruptive activity since the 1970s, attributed to magma replenishment and mixing at depth[13–19]. The geochemistry of eruption products implies arrival of a primitive magma rich in volatiles and alkalis and of radiogenic-Sr character, heralding increased eruption frequency, magma output rate, explosivity and seismicity[13–19]. The appearance of magma with distinct source characteristics was first proposed from the elemental and isotopic chemistry of erupted lavas[13,20,21] and later by study of melt inclusion, mineral and bulk geochemical data, as well as numerical modelling and volcanic gas monitoring[14–19,22–24]. However, the exact intrusion trajectory of the new magma through the plumbing system, a complex network of variably connected sill-like storage regions and dykes, remains poorly constrained. Magmas eventually erupt at the summit or drain laterally through the flank. All summit and most flank eruptions are fed by the so-called central conduits—the main magma transport system—but eccentric flank eruptions can tap into deep, undegassed magmas through pathways that bypass the central conduits[14,19,23,25].

Early mineral phases hold key information on the architecture of magma plumbing systems. At Mt. Etna, typical products are porphyritic trachybasalts with varying fractions of mm-sized plagioclase, olivine and clinopyroxene crystals all showing growth zoning in response to polybaric crystallisation of the hydrous magma along a vertically extended, highly dynamic plumbing system[26–32]. Plagioclase textures and compositions record shallow crystallisation related to degassing (<12 km[27]). Zoned olivines reflect distinct magma-mush pockets within Etna's shallow plumbing system (<12 km[29–31]), only rare Mg-rich compositions informing about deeper (≥20 km) parts of the edifice[31,32]. Clinopyroxene potentially holds a unique further record of magma history, firstly, because it grows across the entire crustal column[17,26,28], and secondly, because elemental diffusion is relatively slow[33,34], potentially preserving the history of protracted processes[35]. In this contribution, we used laser ablation-inductively coupled plasma mass spectrometry (LA-ICPMS) to map trace elemental distributions within clinopyroxene crystals from recent (1974–2014) eruptions at Mt. Etna (Supplementary Tables 1 and 2). Many trace elements are little affected by changes in temperature, pressure and $H_2O$ content relative to major elements[36], and transition metals can record mafic recharge events as zones of enrichment[37].

We combine analysis of the distribution of high-Cr zones in clinopyroxene with thermobarometry to track the appearance and movement of primitive magma in the plumbing system. Our data provide new insights into the links between different areas of the magma reservoir, the timing of magma migration and the association of a new intrusion with increasing eruptive activity over the past four decades.

## Results

**Clinopyroxene zoning.** Although quite homogeneous in major elements, the new trace element maps reveal that Mt. Etna clinopyroxene (titanoaugite) preserves very sharp zoning in the composition of several trace elements, particularly Cr (Fig. 1). Zones rich in Cr are also enriched in other compatible metals (e.g. Ni and Sc) and are relatively poor in incompatible elements such as La, Nd, Zr and Nb (Fig. 1), reflecting growth after arrival of a new primitive magma into the crystallising reservoir[38]. Chromium maps show the strongest zonation and concentration contrasts. This is likely the result of two factors. Firstly, because Cr is compatible in the clinopyroxene lattice[39]. More importantly, however, rapid crystallisation of titanomagnetite and clinopyroxene deplete[39] the hybrid magma in Cr soon after recharge, leading to the prevalent growth of zones low in Cr abundance. Titanomagnetite is a common inclusion in Etnean clinopyroxenes[22,40], where it is typically spatially associated with zones of Cr-enrichment (Fig. 1), supporting the notion of rapid Cr fractionation after mafic recharge. The contrast in Ni concentration in clinopyroxene is less pronounced. This is largely because the hybrid magma is less effectively depleted in Ni due to limited olivine crystallisation (typically restricted to shallow parts of the plumbing system[29,30]).

Chromium-rich zones up to $10^3$ ppm mostly occur near the rims of clinopyroxenes, suggesting growth shortly before eruption[6]. This finding indicates a close link between mafic recharge, magma mixing and volcanic eruption. Maps of glomerocrysts often show Cr-rich layers overgrowing entire aggregates of crystals (Supplementary Fig. 1), from which it can be inferred that the glomerocrysts were assembled prior to the recharge event. Chromium-rich rims are relatively enriched in Mg and depleted in Fe (Fig. 2; Supplementary Fig. 2), as expected from mafic replenishment[38]. The magnitude of the enrichment, however, is much larger in Cr, producing sharper oscillations (Fig. 2; Supplementary Fig. 2). Using clinopyroxene/melt partition coefficients from Etnean hawaiites[39], it can be calculated that the melts in equilibrium with Cr-poor cores and Cr-rich rims were distinctly different (Supplementary Fig. 3; Supplementary Table 3). The calculated increase in transition metal concentration and concomitant decrease in rare earth and high-field strength element content are consistent with the arrival of a new primitive magma into the system.

Chromium-rich rims are euhedral and nearly always enclosed within a Cr-poor outermost rim, where Mg and Si decrease and Al and Ti increase sharply (Fig. 2; Supplementary Fig. 2). Aluminium incorporation into the tetrahedral site, accompanied by Ti for charge balance, increases significantly with magma cooling rate. This has been shown in experiments crystallising synthetic Etna basaltic melts[41,42] and empirically, both at the chilled margins of dykes and lava flows (ref. [41] and references therein) and in outermost crystal zones formed upon ascent and emplacement of alkaline magmas[43]. It follows that Cr-poor outermost rims likely record final magma decompression and surface crystallisation.

Clinopyroxene cores are often rich in Na (Supplementary Fig. 2), a typical feature of crystallisation at depth[44]. The Na-rich cores are subhedral to anhedral, often corroded and show complex zoning patterns, particularly in mildly incompatible elements such as Zr and Nd (Fig. 1). This implies partial resorption of previous (antecryst) cores upon arrival of hotter magma. Importantly, the recycled cores are frequently oscillatory zoned in Cr but at concentrations one order of magnitude lower than the rims (Supplementary Fig. 4). This zonation likely also reflects periodic recharge but with relatively evolved magma, not highly enriched in Cr. This type of recharge can readily be distinguished from the more primitive magma intrusion that gave

rise to dramatic Cr-enrichments and triggered magma ascent and eruption. We refer to this primitive magma as the 'mafic intrusion'.

**Tracking the intrusion of mafic magma through time.** The 1974 eccentric eruption, which built the Mount De Fiore scoria cones and associated lava fields, represents the first surface expression of the arrival of new primitive magma considered responsible for the on-going era of sustained vigorous volcanic activity[13–15,19,23]. Significantly, in the 1974 eccentric eruption products, 93% of the analysed crystals show the Cr-enrichment (Fig. 3; Supplementary Table 4). High-Cr zones occur at the rim of large antecrysts and glomerocrysts, or in the core of small crystals that likely represent phenocrysts from the final erupted magma (Supplementary Fig. 1). Based on location and chronology of earthquakes[45], the 1974 eruption was previously explained with deep tectonic fracturing[23], but the new mineral maps unequivocally highlight the role of the mafic intrusion. The compositions of erupted lavas[23] plot as hybrids between the compositional melt end-members

calculated from the chemistries of distinctive low- and high-Cr zones in clinopyroxene (Supplementary Fig. 3b). The eccentric 1974 products plot towards the Cr-rich end-member, whereas coeval summit (central conduit) lavas plot nearer the Cr-poor end-member. Rubidium and Ba were mapped in selected crystals to test whether the inferred mafic intrusion was alkaline[13,23] but no zonation was found, concentrations being close to detection limits (Supplementary Fig. 5).

From the 1974 lavas, we extended clinopyroxene mapping to eruption products as young as 2014 (Fig. 3), working on the premise that Cr-rich growth zones trace arrival of batches of mafic intrusion and permit the reconstruction of its invasion through the plumbing system. Many crystals from the eccentric eruptions in 2001 (82%) and 2002–2003 (68%) show Cr-enrichments equivalent to those found in 1974 crystals, either at antecryst rims or in phenocryst cores. Compared to 1974, antecrysts in 2001 and 2002–2003 grew much bigger, and the Cr-enrichments are found over large, variably resorbed cores. This suggests that by this point, the intruding magma was beginning to replenish more mature and crystalline mushes. In central conduit

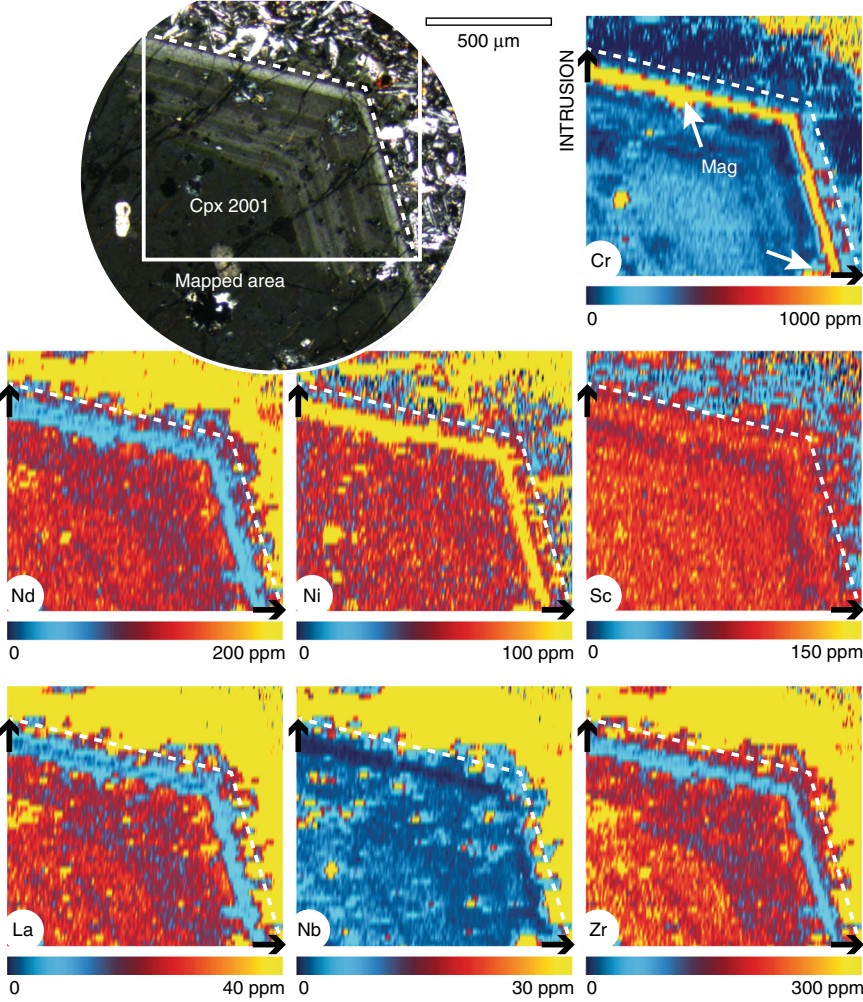

**Fig. 1** Trace element maps of a clinopyroxene antecryst from the 2001 eccentric eruption at Mt. Etna showing primitive magma intrusion prior to eruption. Recharge with primitive magma is recorded as a rim rich in Cr and other compatible metals (Ni, Sc), and depleted in incompatible elements (rare earth La, Nd; high-field strength Zr, Nb). Titanomagnetite (Mag) inclusions are typically associated with Cr-rich clinopyroxene (see white arrows). The maps are quantitative for clinopyroxene and semi-quantitative for other phases (e.g. in the groundmass). Higher concentration levels are indicated by warmer colours on linear scales within the indicated limits. The photomicrograph (transmitted light, crossed polars) provides context, the dashed white line marks the outline of the crystal, and the black arrows mark clinopyroxene growth upon magma intrusion, generating a Cr-rich rim and a Cr-poor outermost rim

lavas, by contrast, Cr-rich antecryst rims and phenocryst cores remained rare in 1992 (13% of clinopyroxene crystals), 1999 (6%) and 2002 (5%), and only started increasing in occurrence from 2013 (14%) and more significantly in 2014 (48%) (Fig. 3; Supplementary Table 4). Thus, with time, the mafic intrusion apparently migrated into the central conduits, consistent with lava chemistries of the intensely eruptive period 2001–2005[46]. Additional evidence of increasing mafic intrusion comes from compositional changes in olivine with time. Although overlapping in composition, the range of forsterite (Mg) contents increased from $Fo_{51-78}$ in pre-1970s products (1669 eruption)[30] to $Fo_{65-83}$ in the period 1991–2008[29], and to $Fo_{70-84}$ in the more recent 2011–2013 lavas[31].

In central conduit lavas, clinopyroxene typically occurs as glomerocrysts (up to 89% of analysed crystals), contrasting with crystals in eccentric eruptions (16–40%; Supplementary Table 4). The ubiquity of glomerocrysts suggests that the central conduits are host to densely populated mushes, which probably form in response to sustained degassing-related crystallisation[27,32,46]. Our data show that the mafic intrusion was very efficient at triggering eccentric eruptions but that penetration of the central mush was slower (Fig. 3) and the arrival of batches of primitive magma not as clearly related to eruption. Nonetheless, the progress of the mafic intrusion inferred from the increasing percentage of crystals with high Cr zones matches the accelerating eruptive activity of the volcano and suggests that the central conduit mush became progressively more permeable to repeated injections of the new magma.

In most eruptions, some antecrysts also show Cr-enrichments in intermediate mantle areas (Fig. 2; Supplementary Fig. 6; Supplementary Table 4). These are likely related to older recharge events that failed to trigger eruption or, alternatively, triggered eruption but failed to drag crystals to the surface. Injections of mafic magma could additionally have formed ribbons in blended ambient magma, through which suspended crystals then migrated[5,47], producing double (or triple) Cr-enrichments (Fig. 2; Supplementary Fig. 6). Where intermediate zones are corroded, we favour the hypothesis that these crystals could be significantly older witnesses of earlier intrusions, kept in 'cold storage'[11] below

the solidus temperature. Judging from the distribution of Cr-rich zones in intermediate mantle areas of clinopyroxene (e.g. Supplementary Fig. 6), crystal storage was more common in the central conduits, probably because mush mobilisation was delayed.

**Constraints on the depth of magma intrusion.** Clinopyroxene is an early liquidus phase in Etnean magmas and crystallises over a wide range of pressures and high magma water contents[17,26,28,32,40]. Without a barometric context, the new information on Cr-enrichment cannot be interpreted in terms of magma plumbing architecture and dynamics. Accordingly, we applied thermobarometry[44,48] to the crystals that witnessed primitive intrusions to calculate storage depths and reconstruct the appearance and migration of the mafic intrusion through the plumbing system. The resulting crystallisation trajectory (Fig. 4) is polybaric, similar to previous constraints at Mt. Etna[17,26]. The results confirm that on average, antecryst cores formed at deeper levels (300–900 ± 200 MPa and 1163–1216 ± 25 °C) than Cr-rich rims (200–600 ± 200 MPa and 1148–1200 ± 25 °C). Pressure estimates for central conduit 2014 clinopyroxene are somewhat lower than eccentric 1974 and 2002–2003 eruptions, however, data overlap within uncertainty.

The steep decompression trajectory (Fig. 4) highlights the efficiency of the mafic intrusion in mush mobilisation and eruption triggering. The recharge pathways are consistent throughout 1974–2014 and are in agreement with GPS, seismic and other geophysical data collected over the last decade[49]. Antecryst cores were entrained as magma travelled from mantle depths through the lower and intermediate crust. Deep intrusion and mush-recycling is consistent with volcano-tectonic seismicity occurring at 10–30 km beneath the western and southern sectors of Mt. Etna, which has been interpreted as episodic magma recharge from depth[49].

Crystallisation of clinopyroxene in response to the arrival of mafic magma occurred at depths matching the location of a major aseismic high P-wave velocity body (HVB) detected at 3–18 km[50–52]. This body is the most important feature revealed by

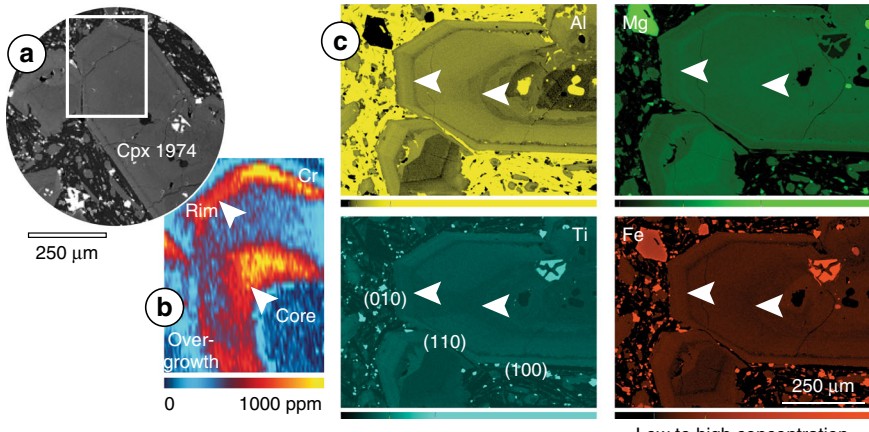

**Fig. 2** Comparison between Cr zoning and major element zoning on a clinopyroxene antecryst from the 1974 eccentric eruption at Mt. Etna. **a** Backscattered electron (BSE) image provides context; the white rectangle marks the location of the Cr map. **b** Quantitative map of Cr obtained by LA-ICPMS. **c** Semi-quantitative maps of Al, Ti, Mg and Fe obtained by FE-SEM–EDS. The crystal shows two Cr-rich zones (see white arrows), which could represent two recharge events or alternatively swirling of magma and migration of crystals following complex pathways[5] after one recharge event. The Cr-rich rim is relatively rich in Mg and poor in Fe, but Mg–Fe zoning is weak in the Cr-rich mantle, probably due to diffusive relaxation with storage time[66,67]. Aluminium zoning is sharp, as expected from its slow diffusion[64,65], and the decoupling between Al and Cr rules out kinetic controls on Cr zoning. The outermost rim of the crystal is defined by a decrease in Cr–Mg and an increase in Al–Ti, as expected from crystallisation under increasing magma cooling rates[41,42], here linked to final magma ascent and degassing. There is a minor sector zoning effect on (010), slightly depleted in Ti–Al but with general concentrations in major and trace elements similar to other sectors in the map

seismic tomography and interpreted as a solidified intrusive complex beneath Mt. Etna[50–52]. Recent seismic and deformation data indicate that magmas prevalently rise along the western margin of the HVB and stagnate within and above it, following discontinuities in crustal lithology[49–53]. Apparent depths of crystallisation of Cr-rich rims are consistent with the prominent ~10 km intra-crustal boundary marking the top of the crystalline basement and the base of the carbonate platform (Fig. 4). Magma storage and periodic magma mixing have previously been pinpointed to similar depths with melt inclusion (10 ± 2 km[14,15,23]) and mineral data (e.g. <500 MPa[32] ~<16 km; 5–8 km[28]).

Finally, we consider the significance of Cr-poor outermost rims. The most obvious explanation is that they crystallised upon final ascent from the reservoir to the surface (Fig. 4). An alternative explanation would be crystallisation at ~10 km prior to magma mobilisation and eruption. However, Cr-rich rims, Cr-poor outermost rims and groundmass microcrysts define a common evolutionary trend in terms of major element contents, where Al and Ti increase sharply with the decrease in Mg# and Si (Supplementary Fig. 2). This suggests continuous polybaric fractionation of magma under increasing undercooling conditions[41–43]. Therefore, it is likely that Cr-poor overgrowths crystallised with final decompression and degassing, within the shallow crystallisation depth range of olivine and plagioclase (Fig. 4).

**Timescales from intrusion to eruption**. There is a consistent thickness of Cr-rich rims and Cr-poor outermost rims (53 ± 32

μm and 63 ± 45 μm, respectively, in crystals from eccentric eruptions, accounting for ~11% and 12% of crystal growth). The consistent growth pattern in outermost zones of clinopyroxenes from eruption products spanning 40 years (Supplementary Table 4) may be relevant if the crystallisation of the euhedral rims relates to the time elapsed between recharge, magma mobilisation and eruption. Whereas the bulk of the antecrysts must have grown episodically over protracted timescales, dominated by long periods of storage without growth[11], the following analysis works from the reasonable assumption that upon arrival of the eruption-triggering batch of mafic intrusion, crystal growth was continuous.

Etnean clinopyroxene growth rates have been estimated empirically[26,54] and experimentally[55], converging at a figure on the order of $10^{-8}$ cm/s at low degrees of undercooling (<10–20 ° C). According to this growth rate, Cr-rich rims are inferred to have formed over ~6 days, reflecting the time elapsed between intrusion and mobilisation (Fig. 5). Subsequently, Cr-poor outermost rims are inferred to have formed over ~7 days, during magma ascent to the surface (Fig. 5). The accuracy of these calculations obviously depends on the veracity of growth rates. It is noted that degassing and high undercooling at shallow levels accelerate crystal growth (for example, ref. [56]) and therefore the timescales calculated from the thickness of the outermost rims should be considered maxima. Experimental evidence shows that growth rates of polyhedral crystals can vary by up to one order of magnitude with magma undercooling[57]. Accordingly, an increase in growth rate up to $10^{-7}$ cm/s would translate into minimum timescales for magma ascent of 0.7 days. Higher undercooling would produce clinopyroxene with hopper to dendritic

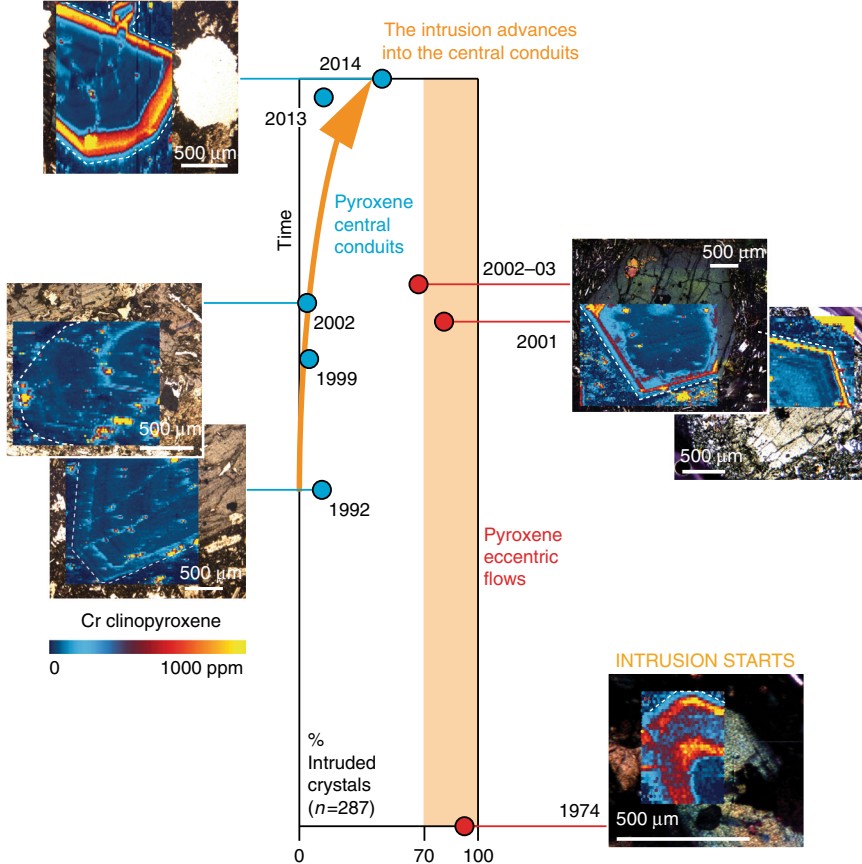

**Fig. 3** Timeline of intrusion of Mt. Etna plumbing system with primitive magma recognised as zones enriched up to $10^3$ ppm Cr in clinopyroxene crystals. In eccentric eruptions (red circles; right column), at least ~70% of crystals record magma intrusion. In central conduit eruptions (blue circles; left column), intrusion was minor in the 1990s and 2000s but is increasing in the present decade, correlating with the increase in eruptive activity. 'Intruded crystals' include antecrysts with Cr-rich rims and phenocrysts with Cr-rich cores (Supplementary Table 4; Supplementary Data 1)

morphologies[41,57], which are not observed in these Etnean lavas. It is also noted that magma temperature during transport could drop below the clinopyroxene solidus, causing the opposite effect. The euhedral shape of rims and outermost rims supports continuous crystallisation at relatively low undercooling, without further resorption.

If the clinopyroxene growth timescale estimates are accurate, they imply maximum recharge to eruption timescales of only 2 weeks (Fig. 5). Similarly short recharge-to-eruption timescales were calculated independently with olivine diffusion chronometry[29] for eccentric eruptions and mafic injections recorded by forsterite-enrichments. Increased $CO_2$ flux in the days prior to the onset of eruptive activity in 2006[24] also supports rapid evacuation of magma upon mafic intrusion. These geochemical indicators for short timescales are supported by geophysical evidence (Fig. 6). For example, the significant 1974 seismic crisis on Mt. Etna started only 10 days before the eccentric eruption, the highest seismic energy release and the deepest earthquake being registered, respectively, 9 and 6 days before eruption onset[23,45] (Fig. 6). The 2001 eruption was preceded by months of gradual pressure build-up in the shallow plumbing system[14], and

heralded by 5 days of very intense seismicity, ground deformation and fracturing[14,22,25]. The somewhat longer timescales obtained for the 2002–2003 eruption (Fig. 5) are in line with the intense pre-eruptive micro-seismicity recorded <2 months prior to eruption and interpreted as precursory deep magma input[58].

Considering that Cr-rich clinopyroxene crystallised within the main magma storage zone (Fig. 4) at ~10 km depth (Fig. 4), 7 days of travel imply an ascent rate of 60 m/h. At the high-undercooling end, minimum growth timescales of 0.7 days would translate into ascent rates of 600 m/h. Considering also the uncertainty in thickness measurements (Supplementary Table 4), the clinopyroxene data provide a range of ascent rates of 35–2080 m/h, from mush mobilisation at the main reservoir to eccentric eruption. Previous estimates at Etna vary widely but indicate fastest average ascent at shallow depths <4 km due to common $H_2O$ degassing[26,59], on the order of 140–1400 m/h[60], 720–2880 m/h[55] and 36–1116 m/h[40], also in agreement with other volcanic systems around the world (ref. [40] and references therein). Average mantle-to-surface ascent is typically slower (1–10 m/h)[26,61] and probably reflects periods of stagnation in the crust[62]. Our new data highlight that injections of the mafic intrusion are capable of

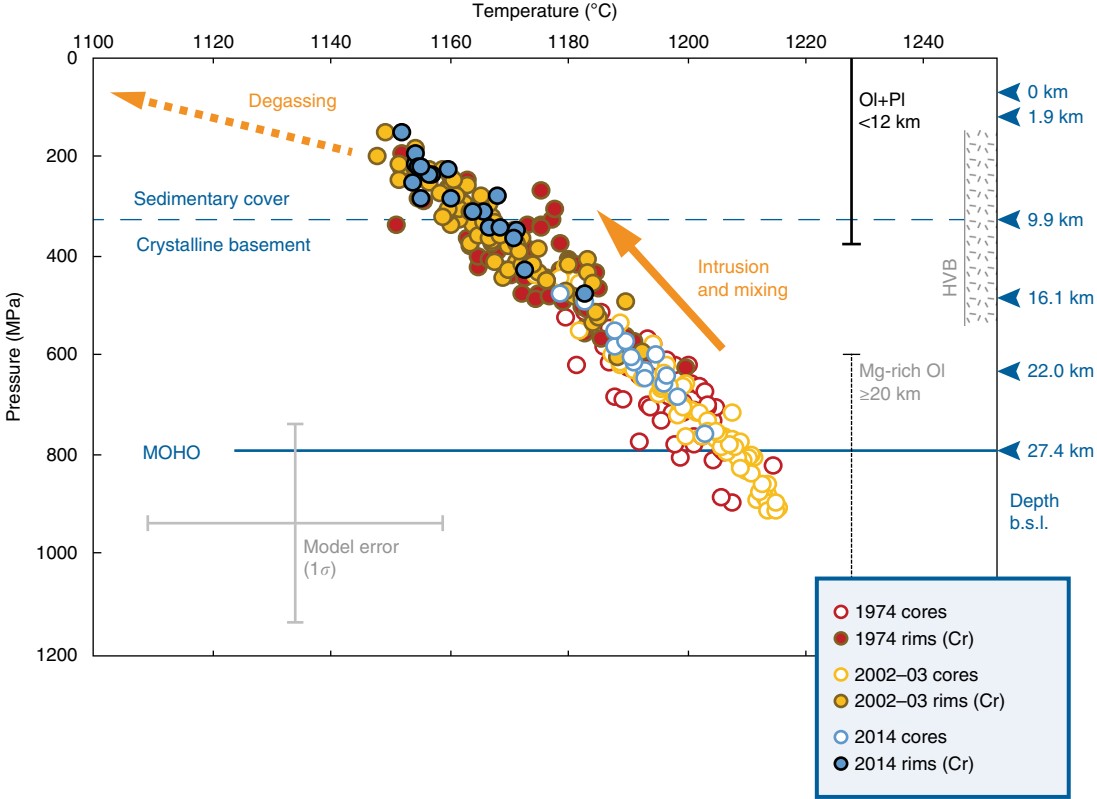

**Fig. 4** Clinopyroxene thermobarometry and crustal context below Mt. Etna. Clinopyroxene cores and Cr-rich rims define a polybaric crystallisation trend, where cores that crystallised at 300–900 ± 200 MPa and 1163–1216 ± 25 °C were sampled by intruding magmas from which Cr-rich rims crystallised at 200–600 ± 200 MPa and 1148–1200 ± 25 °C. Input data for P–T calculations are clinopyroxene compositions from the 1974 and 2002–2003 eccentric eruption and the 2014 central conduit eruption (Supplementary Fig. 2; Supplementary Data 2), and a representative bulk crystal-poor scoria of the 1974 eccentric eruption[23]. Temperatures were estimated using the pressure-independent, melt-dependent calibration of ref. [76]. Pressures were estimated using the melt-independent, $H_2O$-in-melt-dependent, temperature-dependent calibration 32b of ref. [44]. We assumed a conservative $H_2O$ content in the melt of 2 wt%[17,26]. Depth equivalences of major lithological boundaries are indicated across the crustal stratigraphy[78]: 3 km a.s.l. to 0 km: Volcanics and postorogenic clays, 2.47 g/cm³; 0–1.9 km: Marly and argillaceous–arenaceous flysch, 2.57 g/cm³; 1.9–9.9 km: Carbonate rocks, 2.61 g/cm³; 9.9–16.1 km: Granitoids and metamorphic quartzites, 2.66 g/cm³; 16.1–22 km: Felsic granulites, 2.70 g/cm³; 22 km to 27.4 km (Moho): Mafic granulites, 3.03 g/cm³. The Moho depth is after ref. [79]. The depth range of the high P-wave velocity crustal body (HVB; 3–18 km) is after ref. [52]. The HVB is interpreted as a fossilised plexus of sills and dykes related to past volcanism. At present, the ascent of magma is channelled along the western border of the HVB[49-52]. The shallow portion of the plumbing system is composed of multi-level, interconnected mushes where clinopyroxene continues to crystallise together with olivine and plagioclase (<12 km, largely controlled by magma degassing[27-32]). Mg-rich olivine (and clinopyroxene) compositions are sampled more rarely by erupted lavas and are inferred to crystallise at 20 km or deeper[31,32]

evacuating and transporting magma rapidly, particularly through eccentric eruptions.

## Discussion

Our new analysis shows that Cr zoning in clinopyroxene provides an excellent record of mafic replenishment and magma mixing. Chromium partitions preferentially into clinopyroxene[39], where chemical diffusion is relatively slow[33,34], enabling preservation of protracted magma histories. Upon mafic recharge into resident mushes, high-Cr clinopyroxene crystallises together with titanomagnetite, driving down the Cr content of the hybrid magma sharply[39] and producing pronounced oscillatory zoning (Fig. 1). This explanation for sharp zonation in Cr is supported by petrological modelling with Rhyolite-MELTS[63] showing that the 1974 eccentric magma[23] with 2 wt% $H_2O$ at 200–600 MPa fractionates titanomagnetite and clinopyroxene. Changes in Cr concentration show overall correlations with Mg/Fe; however, when Cr-enrichments occur in the intermediate mantle of the crystal, the Mg–Fe zoning is weak or absent (Fig. 2; Supplementary Fig. 2). Divalent Mg and Fe diffuse faster than trivalent

cations[64,65] and Mg–Fe zoning can become reset by diffusion and annealing in the course of crystal storage (e.g. at Taupo Volcanic Zone, New Zealand[66,67]). Trivalent Cr and Al, by contrast, preserve zoning across entire crystals (Fig. 2). Importantly, Cr-enrichments do not correlate with enrichments in other elements that diffuse slowly. For example, Cr-rich zones are poor in Al (Fig. 2; Supplementary Fig. 2). This indicates that episodes of high Cr growth are not due to local enrichment in slow diffusing elements at the liquid boundary layer surrounding rapidly growing clinopyroxene, as has been suggested in other volcanic systems (e.g. at Hawaii[68]). In plutonic clinopyroxene, Mg–Cr-enrichments have been interpreted to reflect local cooling and oxidation of the magma[47]. By analysing the Mt. Etna crystals for a range of trace elements, we established positive correlations between Cr and other transition metals and negative correlations with incompatible elements (Fig. 1; Supplementary Fig. 3), supporting the view that Cr-enrichments represent arrival of mafic magma into the plumbing system.

The clinopyroxene record provides new constraints on the pathways and timescales of intrusion of fresh primitive magma into

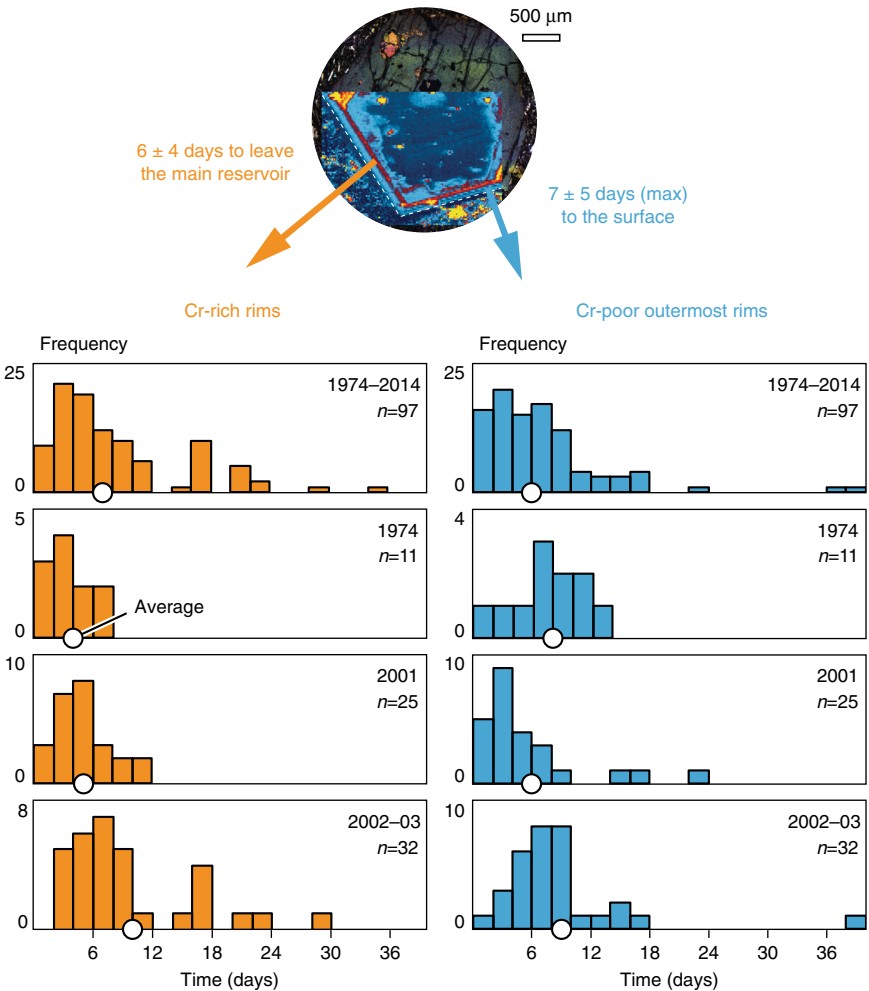

**Fig. 5** Intrusion to eruption timescales calculated using the thickness of Cr-rich rims and Cr-poor outermost rims. The 'clinopyroxene clock' provides magma intrusion to eruption timescales of ~2 weeks. Timescales of crystallisation were calculated using an average clinopyroxene growth rate of $10^{-8}$ cm/s calculated for Mt. Etna[26,54,55]. Chomium-poor outermost rims potentially grew faster upon final magma ascent and degassing[56], so extracted timescales are considered maxima. The example Cr map is from the 2002–2003 eccentric eruption. The Cr-rich and Cr-poor timescale values on either side of the map are average figures for eccentric eruptions, where most antecrysts show rims crystallised from intruding magma (Fig. 3). The upper histograms include data from all studied eruptions (1974–2014) and the other histograms summarise data from individual eccentric eruptions. The white circles mark the average time value (days) per data set. Note the consistency of timescale results across 40 years of eruptive activity (Supplementary Table 4; Supplementary Data 1)

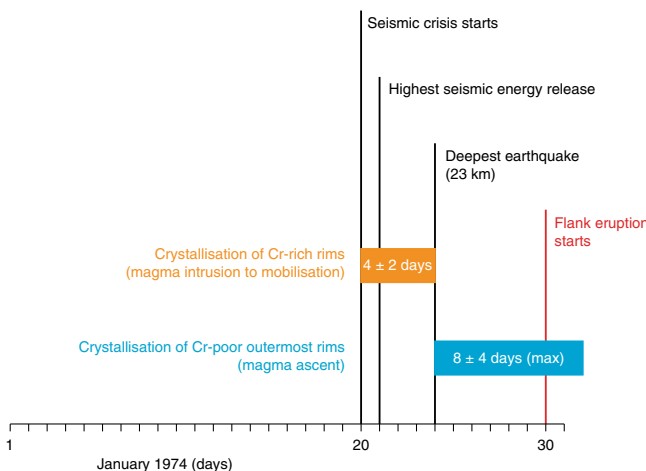

**Fig. 6** Recharge to eruption clinopyroxene timescales compared to pre-eruptive seismic events for the 1974 eccentric eruption. The horizontal axis represents calendar days in January 1974. Major seismic events preceding the onset of eruption[23,45] are marked with vertical lines. Average growth times of clinopyroxene rims and outermost rims for this eruption (Fig. 5) are represented by horizontal rectangles. The 1974 eruption is related to one of the greatest seismic crises ever registered on Mt. Etna. We interpret that seismicity started due to magma intrusion, recorded by the crystallisation of Cr-rich rims. Magma mobilisation fits the timing of the deepest earthquake, and ascent to the surface is recorded by Cr-poor outermost rims

Etna's plumbing system (Fig. 7). It is well established that the feeder system is constantly buffered to the composition of trachybasalt due to continuous supply and limited differentiation of primary magmas in the upper mantle or lowermost crust[26], accompanied by crystallisation of deep Mg-rich olivine and clinopyroxene, which are relatively rarely sampled by Mt. Etna lavas[31,32]. Our data provide evidence of recycling of evolving clinopyroxene mushes in the lower crust, transported by successive replenishments of primitive magma. Geophysical data highlight a preferred path for rising and accumulating magma in recent eruptions. This pathway lies between the western border of the HVB and the eastern border of a shallow low-velocity zone, itself located beneath the volcano's southwestern flank and interpreted as a fractured region favourable for magma rising and ponding[49,50]. Magma storage is inferred at intermediate depth within the HVB ($10 \pm 3$ km[53]; 6–15 km[51]; 8–9 km[49]) and at shallow depth ($3 \pm 2$ km[53]; 0–3 km[50]; 3–5 km[51]; 4–5 and 2 km[49]). Our results indicate that Cr-rich clinopyroxene crystallised upon arrival of batches of mafic intrusion at a ~10 km storage region. Where sharp oscillatory zoning in Cr is found near the rims of the crystals, it infers fast evacuation, ascent and eruption upon such mafic recharge. Mixed magma then migrated upwards through the shallow feeder system, composed of a complex succession of interconnected storage zones[27–32]. Where eccentric dykes connect the ~10 km storage region with the surface[23], the magma rose very quickly. Within the multi-level central conduits, magmas ascended through shallow mushy magma pockets[29], which are sustained by magma mixing and degassing[32,46] and dominated by complexly zoned plagioclase[27] as well as olivine[29] and clinopyroxene[28,32]. The crystal cargo of these centrally erupted lavas is more diverse and represents crystals entrained across nearly the entire crustal column.

Migration of the mafic intrusion to the central conduits has become more pronounced with time, on decadal timescales relevant to the population inhabiting Mt. Etna (Fig. 3). This suggests that strong eruptive activity is likely to continue in the near future, in agreement with geophysical predictions[51] and probabilistic modelling[69], because the supply of primitive intruding magma has remained sustained. If the euhedral Cr-rich and Cr-poor zones of the mapped clinopyroxenes grew continuously upon recharge, calculations suggest that crystal growth occurred over days to weeks (Figs. 5 and 6). This rapid growth followed prolonged residence times at low temperatures, found to be on the order of decades to millennia in volcanic systems world-wide[11,12,56]. The new clinopyroxene compositional maps visualise how arrival of fresh mafic intrusion magma effectively triggered eruption, particularly in eccentric events. By contrast, the regular recharges seen as oscillatory zones of much more muted Cr-contrasts (Supplementary Fig. 4) are less effective at tipping the system to eruption[62]. The consistent ~2 week 'trigger' timescale obtained across 40 years of eruptive activity may serve as a clock informing volcano hazard monitoring, as long as the supply of mafic intrusion is sustained. The chief volcanic hazard at Mt. Etna is flank eruption of undegassed magma caused by perturbation of eccentric pathways by the mafic intrusion. Flank lava flows can reach infrastructure and populated areas at relatively low altitudes. Insights from the studied antecrysts suggest that geophysical evidence of activity in eccentric dykes needs to be taken very seriously and that local evacuation efforts may have to work within <2 weeks.

Because clinopyroxene is a common volcanic antecryst, we propose that our line of investigation could be applied to the eruptive products of other active volcanoes. Combined with better constraints on crystal growth rates, such studies have potential to help decode eruption-triggering mechanisms, depths and timescales. Improved constraints on the movement of magma preceding past eruptions could advise future volcano monitoring efforts in relation to the origin of seismic or deformation signals and the time available for hazard evaluation and emergency planning.

## Methods
**Crystal database**. We analysed 287 clinopyroxene crystals in Mt. Etna lavas spanning 40 years of eruptive activity (1974–2014; Supplementary Fig. 7; Supplementary Tables 1 and 4; Supplementary Data 1). We collected samples from eight eruptions (1974, 1992, 1999, 2001 lower vents, 2002 north flank, 2002–2003 south flank, 2013, 2014) in two field campaigns in 2015. We studied 30 μm polished thin sections of 15 samples with petrography, LA-ICPMS quantitative trace element mapping[37] and field emission scanning electron microscope (FE-SEM) imaging, major element EDS analysis and mapping. We carried out LA-ICPMS maps of up to five crystals per thin section and to derive population statistics, obtained LA-ICPMS transects of between 15 and 44 crystals per thin section depending on rock crystallinity. To account for potential variability of zoning occurrence and thickness, we carried out core to rim transects perpendicular to different crystal faces, up to five per crystal depending on crystal size and shape.

**LA-ICPMS measurements**. Laser ablation-inductively coupled plasma mass spectrometry data were obtained in 2015 and 2016 at the Geochemistry Laboratories of Trinity College Dublin. We used two separate instruments, both equipped with a 193 nm Excimer UV ArF laser with a Helex 2-volume ablation cell, and a quadruple ICPMS mass spectrometer. The first unit is a Teledyne Photon Machines G2 laser coupled to a Thermo Instruments iCapQs ICPMS. The second is a Teledyne Photon Machines Excite coupled to an Agilent 7900. In both cases, ablation was performed in ultra-pure He to which Ar make-up gas with a trace amount of $N_2$ was added for efficient transport and to aid ionisation. Details of laser parameters, gas flows and mass spectrometer operation are given in Supplementary Table 2. Elemental maps were obtained following the method described in ref. [37]. The mapping area was built by overlapping ablation lines to form a rectangular grid. We used a square-shaped laser aperture which was progressively translated by continuous movement of the stage under the fixed ablation site. Spot size selection was determined by the crystal size and set at 24 × 24 μm, 20 × 20 μm, or 12 × 12 μm. Translation speed (measured in μm/s) was set at values of 1.5 times the laser beam size. Repetition rates of 10 Hz were used to ensure overlapping of laser shots across the raster and therefore high spatial resolution, as well as high sensitivity. Individual raster lines were overlapped by 1 μm to avoid un-ablated

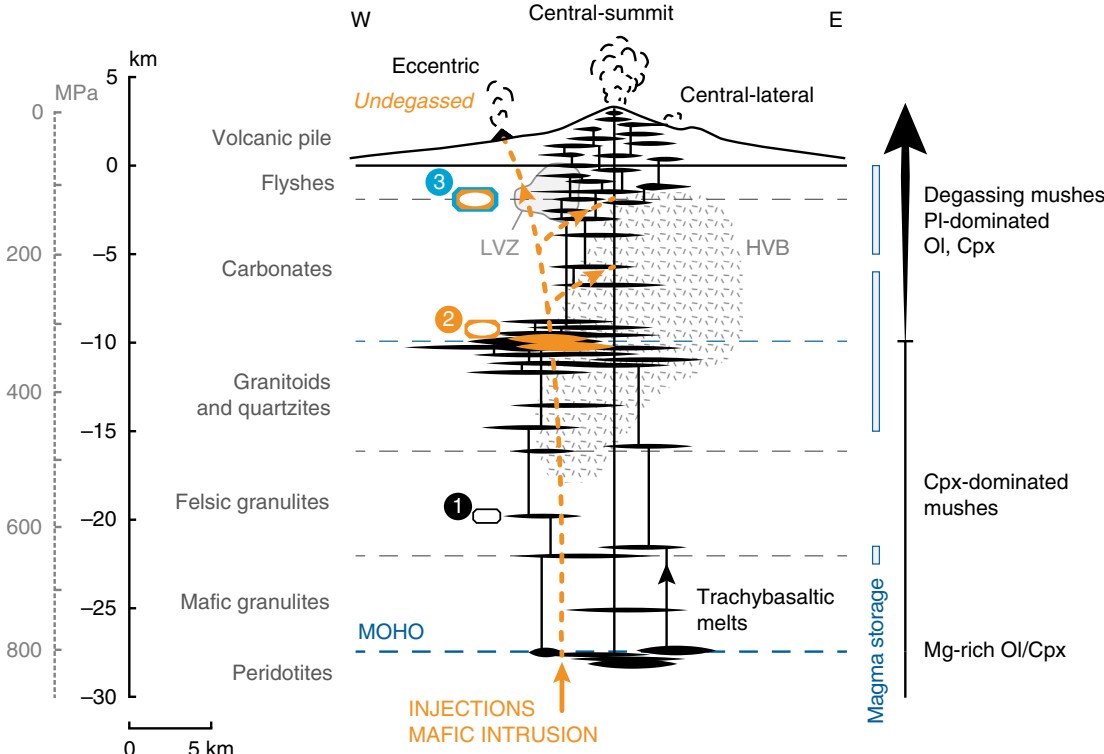

**Fig. 7** Schematic interpretation of the crustal plumbing system feeding Mt. Etna. The transcrustal magma plumbing system is composed of multi-level mush regions, concentrated along stratigraphic boundaries (crustal stratigraphy and rock densities for pressure scale are after refs. [78] and [79]; see also Fig. 4). Our data support a main storage region at ~10 km depth, at the transition between the crystalline basement and the sedimentary sequence. Magma ascent is interpreted between the high-Vp body (HVB; depth range and dimensions after ref. [52]), detected below the summit and eastern Valle del Bove caldera and considered a major solidified plutonic complex, and the shallow low-Vp zone (LVZ)[49–52]. Magma storage depths inferred from geophysical data are represented with vertical blue bars[49–53]. A wide region of low-Vp in the uppermost mantle (<34 km; below the sketch) is interpreted as the source region[52]. Mantle-derived magmas are buffered to trachybasaltic compositions at upper mantle to lower crustal levels, through continuous supply and mixing of primitive melts and crystallisation of Mg-rich olivine and clinopyroxene[26,31,32]. From the storage region at ~10 km and upwards, the system is governed by degassing and mixing in highly dynamic shallow mush pockets[32], dominated by crystallisation of plagioclase[27], olivine[29] and clinopyroxene. Since the 1970s, repeated batches of new mafic magma are intruding into and mixing with resident mushes. This magma has erupted quickly through deep dykes feeding eccentric eruptions and progressively migrated into the shallow, largely crystalline, and degassing central conduits (see yellow dashed pathways). Major and trace element zoning and thermobarometry of clinopyroxene can be explained by (1) recycling of low-mid crustal, relatively evolved mushes; (2) crystallisation of Cr-rich zones (yellow) from intruding magma in the main storage region; and (3) crystallisation of Cr-poor zones (blue) upon final magma ascent and degassing. Progressive intrusion of primitive undegassed magma correlates with increasing eruptive activity in the last 40 years (Fig. 3). We calculate that the time elapsed between replenishment of the main storage system and eruption can be shorter than 2 weeks (Figs. 5 and 6)

gaps in the mapped area that may occur when the true projected laser image is slightly smaller than the nominal aperture size. Elemental transects were built with the same method but using one raster line only. To ensure high spatial resolution and sensitivity in transects, we mostly used a 14 μm spot size (circular mask), with a slow translation speed (10 μm/s) and a high repetition rate (20 Hz). The instruments were tuned with scans on NIST612 glass reference material. Elemental maps were built with Iolite[70] v2.5 in quantitative mode, using NIST610 glass reference material as calibration standard and calcium concentrations obtained by FE-SEM–EDS (see below) as internal standard. Accuracy and precision were monitored using BHVO-2G glass reference material as secondary standard (http://georem.mpch-mainz.gwdg.de/); accuracy was typically better than 10% for Cr, Ga, Ni, Nb, Sc and Zr, and better than 5% for Ba, La, Nd and Sr; precision was typically better than 5% for all analysed elements. Limits of detection[71] were at the sub-ppm level for most analysed elements and typically below 5 ppm for Sc, Cr and Ni.

**Analysis of crystal populations from LA-ICPMS data.** Size and thickness measurements were carried out on spatially registered maps or transects built with the CellSpace[72] module for Iolite. We measured crystal sizes (average of long and short axes), noted if they occurred in single crystals or glomerocryst associations, and noted the presence, location and thickness of Cr-rich zones and their distance to the rim of the crystal (Supplementary Table 4; Supplementary Data 1). The crystallographic orientation of the crystals exposed in the thin section had no noticeable effect on measured thicknesses, nevertheless average thickness values

were obtained for each crystal based on several crystal faces mapped or transected. For intrusion statistics, we considered Cr-rich zones at the rims of antecrysts or at the core of phenocrysts.

**Clinopyroxene clock.** Crystal zone thicknesses were used to calculate timescales of crystallisation (Supplementary Table 4; Supplementary Data 1). Growth of Cr-rich rims was used to estimate timescales from recharge to mobilisation, and growth of Cr-poor outermost rims together with estimates on the depth of clinopyroxene mushes were used to calculate timescales and rates of magma ascent and emplacement. Chromium-rich and Cr-poor zones have polyhedral morphologies typical of relatively low degrees of undercooling[41,57] and show euhedral habits with no evidence of resorption after crystallisation. We used a clinopyroxene growth rate of $10^{-8}$ cm/s[26,54,55] for Cr-rich zones and considered crystallisation up to one order of magnitude faster[57] ($10^{-8}$–$10^{-7}$ cm/s) for Cr-poor zones related to final magma ascent and degassing.

**Extraction of compositions from LA-ICPMS maps.** Average trace element compositions of Cr-rich and Cr-poor clinopyroxene zones were extracted from LA-ICPMS maps and melts in equilibrium with these were calculated with mineral/melt partition coefficients obtained for recent Mt. Etna hawaiites[39]. To extract clinopyroxene compositions from LA-ICPMS maps (Supplementary Table 3), we used Iolite[70] v2.5 to define integrations based on beam intensity. First, we filtered

compositions above a visual calcium threshold (in counts per second) to isolate clinopyroxene compositions from other minerals in the mapped area (e.g. see map of Ca counts per second in Supplementary Fig. 4) and primary and secondary standards measured in the run. Then, we filtered compositions further to those with chromium concentrations between 500 and 2000 ppm for Cr-rich clinopyroxene and between 0 and 400 ppm for Cr-poor clinopyroxene. To avoid including transient signal peaks, we set up a minimum of 2 s of signal complying with the above criteria to include the data in the integrations.

**FE-SEM–EDS measurements**. Major element data were obtained in 2016 and 2017 at the iCRAG labs at Trinity College Dublin. We analysed crystals that record mafic intrusion from the 1974 and 2002–2003 eccentric eruptions and 2014 central conduit eruption (Supplementary Data 2). We used two FE-SEM Tescan instruments: a MIRA XMU equipped with an Oxford X-Max 80 mm$^2$ Energy Dispersive Spectrometer (EDS) detector running Oxford INCA X-ray microanalysis software, and a TIGER MIRA3 equipped with two Oxford X-Max 150 mm$^2$ EDS detectors running Oxford AZtec X-ray microanalysis software. Analyses were performed on carbon-coated thin sections under high vacuum conditions using an accelerating voltage of 20 kV. The Mira instrument was operated at ~400 pA beam current and 18.5 mm working distance; X-rays were acquired for 30 s per spot, reaching >350,000 counts. The Tiger instrument was operated at ~300 pA beam current and 15 mm working distance. X-ray collection was set to stop at $10^6$ counts per spot, which typically took 22 s. Both instruments were calibrated for quantitative analysis with Smithsonian microbeam standards[73] microcline (Si), augite (Al, Ti, Mg and Ca), pyrope (Fe and Mn), anorthoclase (Na and K), and synthetic $Cr_2O_3$ and NiO oxides (Cr and Ni, respectively). Beam current drift was controlled by frequent analysis of cobalt and matrix correction was made using an Oxford-PP (ZAF-type; Z: atomic number, A: absorption, F: fluorescence) procedure. Precision and accuracy were monitored by analysing Smithsonian microbeam standards[73] augite and diopside at the beginning of each analytical session and with sample exchanges. Precision was better than 1% for all analysed elements; accuracy was better than 1% for elements with concentrations above 1 wt% and better than 5% for elements with concentrations below 1 wt%. In addition, the Tiger instrument was used to acquire semi-quantitative maps. The FE-SEM–EDS maps in Fig. 2 have a size of 774 × 580 μm and were built using a pixel size of 0.3 μm, a dwell time per pixel of 5 s and a frame count of 2, with a total experiment time of 14 h.

**Thermobarometry**. Algorithms provided by ref. [48] were applied to major element compositions of crystals that record mafic intrusion from the 1974 and 2002–2003 eccentric eruptions and 2014 central conduit eruption (Supplementary Data 2). We used data from antecryst cores and Cr-rich rims, whereas Cr-poor outermost rims that crystallised upon final magma ascent and degassing have more scattered compositions (Supplementary Fig. 2) and were not included in calculations[74]. We also avoided sector zoned crystals, if present, to eliminate potential kinetic effects on barometry estimates[75]. We chose a representative bulk trachybasalt scoria composition from the eccentric 1974 eruption[23], characterised by low crystal contents, as closest representative of the mafic intrusion melt. Crystallisation temperatures were calculated using pressure-independent thermometry[76], with a relatively low uncertainty of ±25 °C. With regard to pressure estimates, the application of melt-dependent barometers is hindered here by the fact that most clinopyroxene compositions are antecrystic and out of equilibrium with whole rock or groundmass compositions. Melt-independent barometers, by contrast, have recently been used to explore storage depths of magmas feeding ancient (220–100 ka)[77] and recent (2001–2012)[32] Etnean eruptions. To account for the hydrous nature of Etna magmas, we used Equation 32b of ref. [44], which takes into account the water content of the liquid in equilibrium with clinopyroxene (we assumed a conservative 2 wt% average $H_2O$ content[17,26]) and has an uncertainty of ±200 MPa. Solved iteratively using the temperature returned by the thermometer of ref. [76], this barometer has been assessed to provide most accurate results in Etna and other volatile-rich alkaline systems[74,75,77].

**Data availability**. The authors declare that all data supporting the findings of this study are included in this published article and its Supplementary information files. All relevant data are also available on request from the corresponding author (T. U.).

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

## Acknowledgements

This work was funded by Science Foundation Ireland (SFI/12/ERC/E2499 to B.S.K.) and was supplemented with additional funding by The University of Queensland (UQECR1717581 to T.U.) and the Australian Geoscience Council and the Australian Academy of Science (34th IGC Travel Grant to T.U.). We thank Salvatore Ragonesi and Robbie Clarke for help during sampling, David Chew, John Caulfield and Cora McKenna for analytical discussions on LA-ICPMS, Paul Guyett and Thomas Riegler for assistance with FE-SEM–EDS analysis, and Maren Kahl, Silvio Mollo, John Maclennan, Georg Zellmer, Ruadhán Magee and the Magmatic Petrology and Volcanology group at ETH Zurich for discussions on mineral zoning. We acknowledge very insightful suggestions

 

and comments from two anonymous reviewers. We thank the School of Earth and Environmental Sciences at The University of Queensland for covering publication costs.

## Author contributions

The authors jointly designed the research, sampling and experiments, contributed to the discussions and interpretation of the data and participated in the writing stages. B.S.K. obtained the majority of the funding and T.U. led the analytical and data treatment efforts.

## Additional information

**Competing interests:** The authors declare no competing financial interests.

