## [Peer Review File · Nature Communications]

Reviewers' comments:

Reviewer #1 (Remarks to the Author):

The paper proposes that invasion of primitive Cr-rich magma occurred shortly before the 1974 and later eruptions at Etna, giving rise to a distinctive Cr-rich zone in the cpx rims. These distinctive Cr-rich rims dominate crystals in flank ("eccentric") eruptions only and are seen to a much lesser extent in conduit eruptions. This is striking and very interesting. Furthermore, the thickness of the zone is consistent between eruptions. These features have been interpreted by the authors as indicating that each flank eruption was preceded by rapid ascent of primitive magma from depth (the Moho), carrying up cpx remobilised from a mush zone. These magmas take 6 days to be remobilised, based on cpx growth rates, then 6-7 days to ascend, based on Cr-poor rim growth and the same cpx growth rates. The data are interesting, but I am not convinced by the interpretation of these data. There is not enough supporting discussion to rationalize choices of cpx growth rates (this is a huge source of uncertainty), nor to support proposals about magma storage depths. There is not enough critical discussion of alternative mechanisms to generate these Cr-rich zones, including the possibility of diffusive pile-up. There is no clear explanation for why there is a lack of Cr-rich zones in the magmas erupted from the central conduits. This feature suggests to me that these Cr-rich zones grew in very shallow reservoirs beneath Etna. There is no barometry or discussion of other evidence (eg geophysical) to locate regions of mush in the crust. I think that these are interesting data, but much more thought needs to be given as to their origin, which might require a longer paper with more supporting petrological work. Detailed comments below.

Line 35 – intro paragraph. I do not feel the problem is laid out strongly enough into the intro. The ms dives pretty quickly into methodology without describing the question to be answered.

How do the authors rule out diffusive pile-up at the crystal edge for accounting for the Cr-rich zone? How does it compare to other slow diffusers such as Al?

I am not sure about the term "invasion". It is really intrusion.

Line 106 – does this clinopyroxene growth rate account for a temperature increase in the melt as the primitive magma intrudes? Presumably, to allow continued growth of the crystal, the temp is still below the solidus for cpx, but less so, so growth rate would slow? What constraints can you place on temperature? I would like to see a phase diagram for Etna.

Line 108 – undercooling would be much larger during ascent and degassing, as solidus rises. Does this calculation of 7 days use the same mean cpx growth rate as the above calculation? If so this cannot be correct.

Line 109- what is an "incubation period" in this context?

Figure 3 – rather than use the thickness of the Cr-rich zones as a clock, which requires knowledge of cpx growth rates, why not use the diffusional relaxation of the innermost zone as a clock?

Figure 3 – these timescales on the right seem very long for ascent times for basalt – what is the rate? The authors propose these are timescales for ascent from the Moho. What is the evidence for this?

What depth do the authors envision for the intruding primitive magma? Why are the central conduit magmas not affected?

Line 124 – now I am confused. I thought that the idea was that shallow mushes were being invaded by primitive magma, causing the Cr-rich rim. Now there is a suggestion that the cpx is

coming from the Moho. In this case, it seems even more bizarre that only flank eruptions and not central conduits show the Cr-rich rims. The subtleties of volcano plumbing are insignificant when magmas is ascending to the surface quickly from such depths. How can you rule out mush disaggregation in much shallower magma reservoirs in the crust?

Can you make estimates of the pressure of crystal growth using cpx-melt barometry to constrain storage depth?

Reviewer #2 (Remarks to the Author):

Ubide and Kamber have measured Cr (and other trace elements) in clinopyroxene erupted from Etna over the past ~40 years using laser-ablation ICPMS mapping. They use the presence of high-Cr zones in pyroxene to track the appearance of primitive magma in the reservoir. They find high-Cr zones near the rims of some crystals (a high percentage of the crystals in some eruptions), which they interpret to reflect growth in the presence of primitive magmas within the reservoir, followed by additional crystallization of low-Cr rims due to decompression and ascent. They find that the Cr-rich rims are common in eccentric eruptions early in the current eruptive period, followed by the appearance of Cr-rich rims in centrally-erupted magmas later. They interpret these data to reflect efficient triggering of eruptions by introduction of primitive magmas, and they argue that the timing of the appearance of the Cr-rich rims tracks the migration of the primitive magma from the periphery to the more central areas of the magma reservoir. This work is novel and the potential implications are interesting, and could provide new insights into the links between different areas of a magma reservoir and the timing of magma migration. The ability to track the movement of magma through a reservoir during an eruptive period would be very exciting. However, there are a number of assumptions inherent in their interpretation that I think require a bit more discussion in the manuscript.

For example, I am puzzled by the assumption that all of the high-Cr zones near the rims of the crystals in all of the eruptions were caused by the same injection event of primitive magma. The fact that the outermost rims are Cr-poor (which the authors argue is due to rapid Cr depletion of the melts during crystallization) would suggest that the high Cr contents in the primitive magmas are very susceptible to modification and won't survive long in the shallow reservoir. Given this, how is it that the same magma that caused the high Cr rims in the 1974 clinopyroxene was responsible for eventually causing high Cr zones in the 2014 clinopyroxene? Are the authors envisioning some form of armoring of a channel such that the first magma into a region leads to high Cr rims and triggers an eruption, but that subsequent parts of the same magma body passing through that area would maintain their high Cr contents until they react with other regions? Or are they suggesting that multiple batches of high Cr magma entered the system at different times and in different locations? The authors start with "the premise that Cr-rich growth zones trace the primitive magma and permit reconstruction of the invasion through the plumbing system" which seems to imply that they consider the first option to be the case, but they don't discuss how the

high Cr contents would be maintained during this process.

In addition, they make the argument that the location of the high-Cr zones close to (but not at) the rims of the crystals implies a consistent time between injection and crystallization of high Cr cpx, growth of Cr-poor rims during decompression, and eruption. However, at least one of the grains that they show in Fig 2 (and some that they discuss in the methods supplement) shows two zones of equally high Cr in the 1974 eruption. Why didn't the first invasion trigger an eruption? If the high-Cr zones aren't always triggering eruptions, then it seems that the arguments about the ~6 day interval between the formation of the high-Cr zone and eruption (based on the thickness of the low-Cr rims) would not always be valid. Maybe some of the low-Cr rims took much longer to grow below the surface, rather than resulting from rapid crystallization during ascent.

In addition, the magma ascent rate calculations in the manuscript are highly dependent on the model of decompression crystallization causing crystallization of the Cr-poor rims (and resulting choice of growth rates). Crystal growth rates in general are very poorly constrained, and can vary by many orders of magnitude depending on the circumstances of the growth. So while it is plausible that the rims could have crystallized during ascent, it is certainly not required and the growth rates could be very different from assumed.

Other more minor points:

Line 13 (and elsewhere): They mention at several places throughout the manuscript "statistical analysis" of the data. What in particular does this mean - what kind of statistical analysis was performed? Does this simply refer to calculations of the percentage of crystals containing a high-Cr zone for each eruption?

Line 22: "establishing a new approach to volcanic hazard monitoring and prediction". While this method does potentially provide new insights into the triggering of eruptions and the movement of magma through the reservoir, I don't think it is directly applicable to monitoring and prediction, since the eruption has to have happened in order to measure Cr in the pyroxene crystals.

Line 39-41 and supplementary methods: I would have liked to see more information about how the maps were constructed rather than simply citing a reference in the supplement - they don't need to reproduce everything, but some information about how the spacing of the laser spots (or the rasters), and whether any interpolation was done between volumes actually analyzed, would have been useful.

Line 54: the absence of olivine inclusions in pyroxene doesn't preclude crystallization of olivine

Line 56: "Consistently, Cr-rich zones up to 10^3 ppm occur..." This sentence implies that Cr-rich zones are ubiquitous, when they make a point of the fact that they are sometimes present and sometimes absent (and even when present are not in all analyzed crystals from a given eruption).

Line 59: as discussed above, I don't think that decompression-induced crystallization during ascent is the only interpretation for these rims.

Lines 106-108: There is much more uncertainty in the growth rates, and therefore the growth times, than is implied by this sentence.

Response to Reviewers

Thank you very much for your constructive and encouraging review of our manuscript entitled “Volcanic crystals as time capsules of eruption history”. We have obtained a large dataset of new major element analyses on clinopyroxene, which we have used to constrain temperatures and pressures of crystallization, allowing us to reconstruct the intrusion of new magma into the plumbing system. We have also expanded the discussion of our data and added supporting evidence in response to all requests. The manuscript is now more comprehensive and better supported.

The following points (i-iv) address the main concerns. A detailed response to all specific comments is given below, including reference to points (i-iv) as well as to relevant sections and lines in the manuscript.

- i. **Clinopyroxene zoning and diffusion:** by obtaining a large dataset of major element compositions on intruded crystals (new Fig. 2, Supplementary Fig. 1, Supplementary Data 2), we have found that slow diffusing elements such as Al do not correlate with Cr. This provides evidence that Cr-rich zones are not related to local enrichment of slow diffusing elements at the liquid boundary layer surrounding rapidly growing crystals, as proposed in other systems (e.g., Hawaii; Welsh et al., 2016). Further, Cr-enrichments correlate with enrichments in Mg, supporting mafic recharge (e.g., Streck, 2008). Mg-enrichments, however, are less pronounced and typically become reset by diffusion and annealing during antecryst storage (e.g., Nakagawa et al., 1999; 2002). We show that Cr zoning is the most accurate recorder of mafic replenishment, and it is preferentially preserved due to slow diffusion.
- ii. **Clinopyroxene thermobarometry and magma storage depths:** we are very grateful for the suggestion of constraining storage depths and temperatures through clinopyroxene thermobarometry (Putirka, 2008) and discussing the results further in the context of crustal stratigraphy and geophysical data (new Fig. 4). Our data define a clear, steep decompression trend. Antecryst cores were recycled by intruding mafic magmas at low-mid crustal levels, and the intruder magma crystallised Cr-rich rims (and Cr-rich phenocryst cores) at the main storage region (ca. 10 km depth, in agreement with the high Vp velocity body imaged by seismic methods; Aloisi et al., 2002; Patanè et al., 2003, 2013). The outermost, Cr-poor rims show significant enrichments in Al-Ti (Supplementary Fig. 1), typical of final magma decompression and degassing (e.g., Mollo et al., 2010, 2013). We combine our new petrological geochemical information on clinopyroxene with previous data on olivine, plagioclase and bulk rocks, as well as geophysical data on the architecture of the plumbing system. Accordingly, we propose an updated, comprehensive reconstruction of the trans-crustal mush column feeding Mt. Etna (new Fig. 7).
- iii. **Clinopyroxene growth rates as a function of magma temperature:** we agree that the choice of growth rates needed further discussion. As indicated by reviewers, if outermost rims crystallised upon final magma ascent to the surface (Supplementary Fig. 1), high undercooling would accelerate crystal growth and therefore the obtained timescale estimates should be considered maxima. Experimental evidence shows that growth rates of polyhedral crystals can vary by up to one order of magnitude with magma undercooling (Kouchi et al., 1983). Cr-rich rims and Cr-poor outermost rims have polyhedral shapes, as opposed to hopper or dendritic shapes that would develop at higher degrees of undercooling and higher growth rates (Kouchi et al., 1983; Mollo et al., 2010; Welsh et al., 2016). We also consider the possibility of crystal dissolution below the solidus temperature, however the rims and outermost rims have euhedral habits. Taking all of these

constraints into account, we now apply growth rates of 10^{-8} cm/s (Armienti et al., 1994, 2013; Orlando et al., 2008) to Cr-rich rims and a range of growth rates of 10^{-8} to 10^{-7} cm/s to Cr-poor overgrowths. Our results are in agreement with diffusion chronometry in olivine (Kahl et al., 2015) and seismic data preceding past eruptions (e.g., Fig. 6). We have calculated magma ascent rates from the main storage zone (ca. 10 km; Fig. 4, 7) to the surface, taking into consideration the range of growth rates and the uncertainty of timescale estimates. Importantly, our results are in agreement with previous estimates at Mt. Etna (Aloisi et al. 2006; Orlando et al. 2008; Mollo et al. 2015) and other active volcanoes world-wide (Mollo et al. 2015 and references therein).

- iv. **Magma intrusion:** we have removed the terms ‘invasion’ and ‘invader’ from the manuscript. We now describe the general process as magma ‘intrusion’ and we use magma ‘injection’, ‘replenishment’, ‘mixing’ etc. for the individual recharge events. We acknowledge that the previous version of the paper was not clear enough regarding the process of intrusion with time, which we envisage to occur through multiple injections of the distinctly primitive intruder magma. Thanks to the new petrological work and barometric estimates, we have been able to constrain the pathways of magma intrusion with time, as we illustrate in the updated architecture of the plumbing system that we propose in Fig. 7.

Comments from Reviewer #1

1. The paper proposes that invasion of primitive Cr-rich magma occurred shortly before the 1974 and later eruptions at Etna, giving rise to a distinctive Cr-rich zone in the cpx rims. These distinctive Cr-rich rims dominate crystals in flank (“eccentric”) eruptions only and are seen to a much lesser extent in conduit eruptions. This is striking and very interesting. Furthermore, the thickness of the zone is consistent between eruptions. These features have been interpreted by the authors as indicating that each flank eruption was preceded by rapid ascent of primitive magma from depth (the Moho), carrying up cpx remobilised from a mush zone. These magmas take 6 days to be remobilised, based on cpx growth rates, then 6-7 days to ascend, based on Cr-poor rim growth and the same cpx growth rates. The data are interesting, but I am not convinced by the interpretation of these data.

Response: Thank you very much, please see points (i-iv) above.

2. There is not enough supporting discussion to rationalize choices of cpx growth rates (this is a huge source of uncertainty), nor to support proposals about magma storage depths.

Response: These issues are now thoroughly discussed in Results sections “Constraints on the depth of magma intrusion” (lines 182-215) and “Timescales from intrusion to eruption” (lines 217-271) of the manuscript. Please see also points (ii-iii) above.

3. There is not enough critical discussion of alternative mechanisms to generate these Cr-rich zones, including the possibility of diffusive pile-up.

Response: This is now discussed in depth in Results section “Clinopyroxene zoning” (lines 76-133). Please see also point (i) above.

4. There is no clear explanation for why there is a lack of Cr-rich zones in the magmas erupted from the central conduits. This feature suggests to me that these Cr-rich zones grew in very shallow reservoirs beneath Etna. There is no barometry or discussion of other evidence (eg geophysical) to locate regions of mush in the crust.

Response: We have obtained >400 major element analyses on clinopyroxene crystals and we have used the new data to constrain temperatures and depths of crystallization, which we have discussed in the context of geophysical data and constraints from other mineral phases and bulk rocks (see Results section “Constraints on the depth of magma intrusion” (lines 182-215), Discussion (lines 273-309) and Figs. 4 and 7). The new data indicate that antecryst cores crystallised at low-mid crustal levels. They were recycled by intruding magmas, which crystallised in the main reservoir at ca. 10 km depth. The intruder magmas were tapped quickly through eccentric eruptions (1974, 2001 and 2002-03). They also intruded the central conduits, but these were more difficult to penetrate, being more crystalline mushes. Regardless, with time, the central conduits became progressively more permeable to the intruder, as shown by the increase of intruded crystals in erupted products through time (Fig. 3), which correlates with the acceleration in eruptive activity. We have now described and discussed our data more clearly and in greater depth in the text and also with new Figs. 4 and 7. Please see also points (ii) and (iv) above.

5. I think that these are interesting data, but much more thought needs to be given as to their origin, which might require a longer paper with more supporting petrological work. Detailed comments below.

Response: Thank you very much, please see points (i-iv) above.

Detailed comments:

6. Line 35 – intro paragraph. I do not feel the problem is laid out strongly enough into the intro. The ms dives pretty quickly into methodology without describing the question to be answered.

Response: We have expanded the introduction significantly and feel that the problem is laid out clearly and strongly from the start in the revised manuscript.

7. How do the authors rule out diffusive pile-up at the crystal edge for accounting for the Cr-rich zone? How does it compare to other slow diffusers such as Al?

Response: Thank you for pointing this out. We have now obtained a large dataset of major element data (Supplementary Fig. 1) and also major element maps (Fig. 2) that clearly show that Cr-rich zones are Al-poor. This indicates that episodes of high Cr growth are not due to local enrichment of slowly diffusing elements at the liquid boundary layer surrounding rapidly growing clinopyroxene, as has been suggested in other volcanic systems (e.g., at Hawaii; Welsch et al., 2016). We now discuss major and trace element zoning in Results section “Clinopyroxene zoning” (lines 76-133). Please see also point (i) above.

8. I am not sure about the term “invasion”. It is really intrusion.

Response: Changed throughout – please see point (iv) above.

9. Line 106 – does this clinopyroxene growth rate account for a temperature increase in the melt as the primitive magma intrudes? Presumably, to allow continued growth of the crystal, the temp is still below the solidus for cpx, but less so, so growth rate would slow? What constraints can you place on temperature? I would like to see a phase diagram for Etna.

Response: According to our new thermobarometric constraints (Fig. 4), the intruding magma recycled clinopyroxene mushes crystallised at $3-9\pm 2$ kbar and $1163-1216\pm 25$ °C. The increase in temperature introduced by the hot recharge translated into dissolution of recycled antecryst cores, which often show resorption features (e.g., Fig. 2). The intruder magma ascended towards the main storage region, where it cooled below the clinopyroxene liquidus, crystallising Cr-rich clinopyroxene ($2-6\pm 2$ kbar and $1148-1200\pm 25$ °C) and magnetite (Fig. 1; corroborated with thermodynamic modelling using Rhyolite-MELTS; Gualda et al., 2012). The polyhedral shape of Cr-rich rims and Cr-poor outermost rims indicate crystallisation at low degrees of undercooling. Under such conditions, empirical and experimental determinations of growth rates at Mt. Etna are on the order of 10^{-8} cm/s (Armienti et al., 1994, 2013; Orlando et al., 2008). However, final crystallisation of Cr-poor overgrowths took place upon final magma ascent and degassing (Supplementary Fig. 1) and growth rates could have accelerated by up to one order of magnitude (Kouchi et al., 1983; Mollo et al., 2010). Hence, we consider timescales of crystallisation of outermost rims as maxima. During transport from the main reservoir to the surface, the intruder magma recycled olivine (e.g., Kahl et al., 2015) and plagioclase (e.g., Giacomoni et al., 2014), particularly when intruding shallow, degassing central conduit mushes. Please see also Results (lines 76-271) and Discussion (lines 273-309) and points (ii-iii) above.

10. Line 108 – undercooling would be much larger during ascent and degassing, as solidus rises. Does this calculation of 7 days use the same mean cpx growth rate as the above calculation? If so this cannot be correct.

Response: Thank you for pointing this out. Please see the response to comment 9 and point (iii) above, as well as Results section “Timescales from intrusion to eruption” (lines 217-271).

11. Line 109- what is an “incubation period” in this context?

Response: Reworded for clarity – please see section “Timescales from intrusion to eruption” (lines 217-271).

12. Figure 3 – rather than use the thickness of the Cr-rich zones as a clock, which requires knowledge of cpx growth rates, why not use the diffusional relaxation of the innermost zone as a clock?

Response: Growth rates and timescale estimates are better and more carefully discussed and supported in the revised manuscript and our results agree with timescales obtained from independent diffusion modelling in olivine, particularly in eccentric eruptions and rims related to mafic recharge (Kahl et al., 2015). Please see Results section “Timescales from intrusion to eruption” (lines 217-271) and point (iii) above.

13. Figure 3 – these timescales on the right seem very long for ascent times for basalt – what is the rate? The authors propose these are timescales for ascent from the Moho. What is the evidence for this?

Response: Based on new barometry estimates, we have calculated ascent rates from the main reservoir to the surface. Our results take into consideration the range of growth rates and the uncertainty of timescale estimates, and are in agreement with previous estimates at Mt. Etna (Aloisi et al. 2006; Orlando et al. 2008; Mollo et al. 2015) and other active volcanoes world-wide (Mollo et al. 2015 and references therein). See also Results section “Timescales from intrusion to eruption” (lines 217-271) and point (iii) above.

14. What depth do the authors envision for the intruding primitive magma? Why are the central conduit magmas not affected?

Response: Please see Results sections “Tracking the intrusion of mafic magma through time” (lines 135-180) and “Constraints on the depth of magma intrusion” (lines 182-215), as well as Discussion (lines 273-309) and points (ii) and (iv) above.

15. Line 124 – now I am confused. I thought that the idea was that shallow mushes were being invaded by primitive magma, causing the Cr-rich rim. Now there is a suggestion that the cpx is coming from the Moho. In this case, it seems even more bizarre that only flank eruptions and not central conduits show the Cr-rich rims. The subtleties of volcano plumbing are insignificant when magmas is ascending to the surface quickly from such depths. How can you rule out mush disaggregation in much shallower magma reservoirs in the crust?

Response: We have rewritten this – please see Results section “Constraints on the depth of magma intrusion” (lines 182-215), Discussion (lines 273-309), Figs. 4 and 7, and points (ii) and (iv) above.

16. Can you make estimates of the pressure of crystal growth using cpx-melt barometry to constrain storage depth?

Response: Thank you for pointing this out. We have applied barometric calibrations to estimate crystallisation pressures and temperatures, and discussed storage depths (see Results section “Constraints on the depth of magma intrusion” (lines 182-215), Discussion (lines 273-309), and point (ii) above).

Comments from Reviewer #2

17. Ubide and Kamber have measured Cr (and other trace elements) in clinopyroxene erupted from Etna over the past ~40 years using laser-ablation ICPMS mapping. They use the presence of high-Cr zones in pyroxene to track the appearance of primitive magma in the reservoir. They find high-Cr zones near the rims of some crystals (a high percentage of the crystals in some eruptions), which they interpret to reflect growth in the presence of primitive magmas within the reservoir, followed by additional crystallization of low-Cr rims due to decompression and ascent. They find that the Cr-rich rims are common in eccentric eruptions early in the current eruptive period, followed by the appearance of Cr-rich rims in centrally-erupted magmas later. They interpret these data to reflect efficient triggering of eruptions by introduction of primitive magmas, and they argue that the timing of the appearance of the Cr-rich rims tracks the migration of the primitive magma from the periphery to the more central areas of the magma reservoir. This work is novel and the potential implications are interesting, and could provide new insights into the links between different areas of a magma reservoir and the timing of magma migration. The ability to track the movement of magma through a reservoir during an eruptive period would be very exciting. However, there are a number of assumptions inherent in their interpretation that I think require a bit more discussion in the manuscript.

Response: Thank you very much, please see points (i-iv) above.

18. For example, I am puzzled by the assumption that all of the high-Cr zones near the rims of the crystals in all of the eruptions were caused by the same injection event of primitive magma. The fact that the outermost rims are Cr-poor (which the authors argue is due to rapid Cr depletion of the melts during crystallization) would suggest that the high Cr contents in the primitive magmas are very susceptible to modification and won't survive long in the shallow reservoir. Given this, how is it that the same magma that caused the high Cr rims in the 1974 clinopyroxene was responsible for eventually causing high Cr zones in the 2014 clinopyroxene? Are the authors envisioning some form of armoring of a channel such that the first magma into a region leads to high Cr rims and triggers an eruption, but that subsequent parts of the same magma body passing through that area would maintain their high Cr contents until they react with other regions? Or are they suggesting that multiple batches of high Cr magma entered the system at different times and in different locations? The authors start with "the premise that Cr-rich growth zones trace the primitive magma and permit reconstruction of the invasion through the plumbing system" which seems to imply that they consider the first option to be the case, but they don't discuss how the high Cr contents would be maintained during this process.

Response: We agree with the reviewer and envision the second scenario, however this was not clear in the previous version of the manuscript. We have now rewritten this part more clearly, provided more supporting evidence (see Results (lines 76-215) and Discussion (lines 273-309)) and illustrated the intrusion process in new Fig. 7. Please see also point (iv) above.

19. In addition, they make the argument that the location of the high-Cr zones close to (but not at) the rims of the crystals implies a consistent time between injection and crystallization of high Cr cpx, growth of Cr-poor rims during decompression, and eruption. However, at least one of the grains

that they show in Fig 2 (and some that they discuss in the methods supplement) shows two zones of equally high Cr in the 1974 eruption. Why didn't the first invasion trigger an eruption? If the high-Cr zones aren't always triggering eruptions, then it seems that the arguments about the ~6 day interval between the formation of the high-Cr zone and eruption (based on the thickness of the low-Cr rims) would not always be valid. Maybe some of the low-Cr rims took much longer to grow below the surface, rather than resulting from rapid crystallization during ascent

Response: We agree this is an interesting feature (Supplementary Fig. 6) and have discussed it more clearly and thoroughly in the new version of the paper (see Results section "Tracking the intrusion of mafic magma through time", specifically lines 172-180). We argue that Cr-enrichments located at crystal mantles are likely related to older recharge events that failed to either trigger eruption or drag crystals to surface, particularly from the central conduits where mush mobilisation was delayed. Alternatively, injections of intruder magma could also have formed ribbons in blended ambient magma, through which suspended crystals then migrated (Barkov & Martin, 2015; Bergantz et al., 2015), producing double (or triple) Cr-enrichments (Supplementary Fig. 6). Where intermediate zones are corroded, we favour the hypothesis that these crystals could be significantly older witnesses of earlier intrusions kept in 'cold storage' (Cooper and Kent, 2014), below the solidus. Thus, for intrusion statistics and timescale estimates (Fig. 3; Supplementary Table 3), we only consider Cr-enrichments located at antecryst rims or phenocryst cores.

20. In addition, the magma ascent rate calculations in the manuscript are highly dependent on the model of decompression crystallization causing crystallization of the Cr-poor rims (and resulting choice of growth rates). Crystal growth rates in general are very poorly constrained, and can vary by many orders of magnitude depending on the circumstances of the growth. So while it is plausible that the rims could have crystallized during ascent, it is certainly not required and the growth rates could be very different from assumed.

Response: Thank you for pointing this out. Please see point (iii) above and responses to comments 9, 12 and 13, and Results section "Timescales from intrusion to eruption" (lines 217-271).

Other more minor points:

21. Line 13 (and elsewhere): They mention at several places throughout the manuscript "statistical analysis" of the data. What in particular does this mean - what kind of statistical analysis was performed? Does this simply refer to calculations of the percentage of crystals containing a high-Cr zone for each eruption?

Response: Yes, our dataset is composed of 287 clinopyroxene crystals from 8 eruptions across 40 years of eruptive activity. We analysed the size, type and zoning pattern of the crystals (Supplementary Data 1). In order to obtain population statistics, we calculated averages of the measurements per year and per type of eruption (Supplementary Fig. 3).

22. Line 22: "establishing a new approach to volcanic hazard monitoring and prediction". While this method does potentially provide new insights into the triggering of eruptions and the movement of

magma through the reservoir, I don't think it is directly applicable to monitoring and prediction, since the eruption has to have happened in order to measure Cr in the pyroxene crystals.

Response: We have removed "prediction" and kept "monitoring".

23. Line 39-41 and supplementary methods: I would have liked to see more information about how the maps were constructed rather than simply citing a reference in the supplement - they don't need to reproduce everything, but some information about how the spacing of the laser spots (or the rasters), and whether any interpolation was done between volumes actually analyzed, would have been useful.

Response: The analytical methods have been expanded, including a detailed description of the mapping technique (see Methods section "LA-ICPMS measurements"; lines 324-352).

24. Line 54: the absence of olivine inclusions in pyroxene doesn't preclude crystallization of olivine.

Response: We agree and have rewritten this part (see Results section "Clinopyroxene zoning"; lines 76-133).

25. Line 56: "Consistently, Cr-rich zones up to 10^3 ppm occur..." This sentence implies that Cr-rich zones are ubiquitous, when they make a point of the fact that they are sometimes present and sometimes absent (and even when present are not in all analyzed crystals from a given eruption).

Response: We agree and have rewritten this sentence (see Results section "Clinopyroxene zoning"; lines 76-133).

26. Line 59: as discussed above, I don't think that decompression-induced crystallization during ascent is the only interpretation for these rims.

Response: With new major element measurements, we have now discussed the origin of the rims in much greater detail – please see Results sections "Clinopyroxene zoning" (lines 76-133) and "Constraints on the depth of magma intrusion" (lines 182-215), as well as points (i-ii) above.

27. Lines 106-108: There is much more uncertainty in the growth rates, and therefore the growth times, than is implied by this sentence.

Response: Thank you for pointing this out. Please see point (iii) above and responses to comments 9, 12 and 13, as well as Results section "Timescales from intrusion to eruption" (lines 217-271).

REVIEWERS' COMMENTS:

Reviewer #1 (Remarks to the Author):

Ubide and Kamber have put together a very thorough and robust rebuttal and the paper is much improved. Recognising and quantifying episodes of magma intrusion and linking them to eruption is an important advance in this field, and reinforces the emerging picture of magma reservoirs as complex, polybaric storage region where magma mixing dominates. I am happy to recommend publication with minor revision. I have just two comments.

1. Line 244: additional evidence for these timescales of intrusion before eruption comes from observation of increased CO₂ flux at Etna prior to eruption (Aiuppa et al., 2007)
2. You might want to discuss paper structure with the editor – some of the “discussion” is in results section at the moment perhaps. The discussion as is is rather brief and reads more like a “conclusion” section.

Reviewer #2 (Remarks to the Author):

Ubide and Kamber have established an interesting data set linking high-Cr zones within the crystals to the influence of primitive magma. I agree that the data show evidence of an increasing influence of a Cr-rich magma in the system over time, and that by looking at changes in the percentage of cpx crystals within a given eruption that show this influence, they see evidence for increasing influence of such a primitive magma in the central conduit compared to the rift zones. In and of itself, this is an important observation and an interesting addition to our understanding of how the magma supply conduits change over time. In addition, by assuming growth rates, they use the widths of the Cr-rich rims and the Cr-poor rims that post-date the Cr-rich growth to estimate ascent rates and times between the growth of the Cr-rich rims and the eruption. It is also intriguing that these growth time scales are similar to the historical records of seismicity preceding eruptions. Overall, I find that the data set is an interesting addition to our understanding of Etna and of magma plumbing systems overall.

I am happy to see that the authors added new barometry to the revisions, as it greatly strengthens their arguments about where the high-Cr rims were growing. In addition, I think that it would also be interesting to compare these with not only the cores of the same crystals (as they did) but also the Cr-poor rims and the crystals that did not contain high-Cr zones. This seems a relatively straightforward way to test the interpretation of the crystals lacking high-Cr zones as being incorporated from a shallow mush, and the interpretation that the Cr-poor outermost rims grew during final ascent. But since this is not the main point of the paper, I don't think it needs to be added prior to publication.

However, there remain two points that I would like to see clarified. First, as both reviewers pointed out in the initial reviews, the duration between influence of primitive magmas and resulting growth or Cr-rich rims and then the time difference between growth or Cr-rich rims and the eruption are highly dependent on the growth rates and on the assumption of continuous, steady crystallization. The first part of this point is now acknowledged in the revised version and the growth rates chosen are justified based on other experimental and observational evidence, but the second assumption is not made explicit. In an overall model of remobilization of mush crystals, variations in growth rate and/or growth hiatuses are required – otherwise, the crystals would be enormous. So while it may be reasonable to argue in this case that growth was continuous once the Cr-rich rims started crystallizing (because it is consistent with seismic swarm timing, etc), it should at least be made explicit that this is an assumption.

Second, I still am not convinced that these data indicate that the Cr-rich magma is a more efficient

eruption trigger than other recharge events. In fact, it seems that this change in percentage of Cr-rich crystals over time is itself an argument against the primitive magma itself triggering the eruptions more effectively than other magmas. The fact that there are eruptions that have widely varying percentages of crystals with the Cr-rich rims indicates that eruptions happen without a strong influence of the Cr-rich magmas. Is there an increase in frequency of eruptions that correlates with the percentage of Cr-rich rims?

Overall, I think that this paper presents some new and interesting data that contribute to our understanding of the magma reservoir at Etna, and more generally to understanding sub-volcanic processes. This paper goes beyond simply presenting some new data in isolation, instead making connections between these data and other information about the volcanic system - in particular, the correspondence between their calculated durations between growth of Cr-rich zones and eruption and the seismic records prior to eruption is intriguing. It is precisely because they are making these connections that I think it's critical to be clear and explicit about which parts of the conclusions are dependent on which assumptions, so that others in the field can accurately incorporate these results into the ongoing discussion in the literature.

Response to Reviewers

Thank you very much for your positive and encouraging review of our manuscript entitled “Volcanic crystals as time capsules of eruption history”. We have addressed all comments, including reorganising Results and Discussion sections to reduce interpretations in the Results, and stressing and clarifying our assumptions. The revised manuscript uses the ‘track changes’ feature.

A detailed response to the comments received is given below, including reference to relevant sections and lines in the manuscript.

Comments from Reviewer #1

Ubide and Kamber have put together a very thorough and robust rebuttal and the paper is much improved. Recognising and quantifying episodes of magma intrusion and linking them to eruption is an important advance in this field, and reinforces the emerging picture of magma reservoirs as complex, polybaric storage region where magma mixing dominates. I am happy to recommend publication with minor revision. I have just two comments.

Response: Thank you very much, please see responses to points 1 and 2 below.

1. Line 244: additional evidence for these timescales of intrusion before eruption comes from observation of increased CO₂ flux at Etna prior to eruption (Aiuppa et al., 2007)

Response: Thank you! We have added and discussed this information (lines 671-673) and also considered gas monitoring data in the introduction (line 67).

2. You might want to discuss paper structure with the editor – some of the “discussion” is in results section at the moment perhaps. The discussion as is is rather brief and reads more like a “conclusion” section.

Response: We have reorganised the Results and Discussion sections to minimise interpretations in the results section, while still keeping the logical flow of the manuscript (see Results and Discussion sections).

Comments from Reviewer #2

Ubide and Kamber have established an interesting data set linking high-Cr zones within the crystals to the influence of primitive magma. I agree that the data show evidence of an increasing influence of a Cr-rich magma in the system over time, and that by looking at changes in the percentage of cpx crystals within a given eruption that show this influence, they see evidence for increasing influence of such a primitive magma in the central conduit compared to the rift zones. In and of itself, this is an important observation and an interesting addition to our understanding of how the magma supply conduits change over time. In addition, by assuming growth rates, they use the widths of the

Cr-rich rims and the Cr-poor rims that post-date the Cr-rich growth to estimate ascent rates and times between the growth of the Cr-rich rims and the eruption. It is also intriguing that these growth time scales are similar to the historical records of seismicity preceding eruptions. Overall, I find that the data set is an interesting addition to our understanding of Etna and of magma plumbing systems overall.

Response: Thank you very much.

I am happy to see that the authors added new barometry to the revisions, as it greatly strengthens their arguments about where the high-Cr rims were growing. In addition, I think that it would also be interesting to compare these with not only the cores of the same crystals (as they did) but also the Cr-poor rims and the crystals that did not contain high-Cr zones. This seems a relatively straightforward way to test the interpretation of the crystals lacking high-Cr zones as being incorporated from a shallow mush, and the interpretation that the Cr-poor outermost rims grew during final ascent. But since this is not the main point of the paper, I don't think it needs to be added prior to publication. However, there remain two points that I would like to see clarified.

Response: Thank you! We agree that thermobarometric constraints on crystals lacking high-Cr zones would be an interesting approach in the future but is out of the scope of the present manuscript. Quantitative constraints on Cr-poor outermost rims would also be useful, however these have more scattered major element compositions due to crystallisation during final magma ascent and emplacement (Supplementary Fig. 2; Supplementary Data 2) and, therefore, do not provide accurate thermobarometric constraints, as observed in other alkaline systems (e.g., ref. 74). This is described in the Methods section (lines 1041-1066). Please see responses to points 3 and 4 below.

3. First, as both reviewers pointed out in the initial reviews, the duration between influence of primitive magmas and resulting growth or Cr-rich rims and then the time difference between growth or Cr-rich rims and the eruption are highly dependent on the growth rates and on the assumption of continuous, steady crystallization. The first part of this point is now acknowledged in the revised version and the growth rates chosen are justified based on other experimental and observational evidence, but the second assumption is not made explicit. In an overall model of remobilization of mush crystals, variations in growth rate and/or growth hiatuses are required - otherwise, the crystals would be enormous. So while it may be reasonable to argue in this case that growth was continuous once the Cr-rich rims started crystallizing (because it is consistent with seismic swarm timing, etc), it should at least be made explicit that this is an assumption.

Response: We agree and have stressed and clarified the assumptions involved in timescale estimates from clinopyroxene zoning, so they are now clear to the reader. We have highlighted both the dependence on growth rates and the assumption of continuous clinopyroxene crystallisation upon recharge (lines 620-623, 630-631, 804-808).

4. Second, I still am not convinced that these data indicate that the Cr-rich magma is a more efficient eruption trigger than other recharge events. In fact, it seems that this change in percentage of Cr-rich crystals over time is itself an argument against the primitive magma itself triggering the

eruptions more effectively than other magmas. The fact that there are eruptions that have widely varying percentages of crystals with the Cr-rich rims indicates that eruptions happen without a strong influence of the Cr-rich magmas. Is there an increase in frequency of eruptions that correlates with the percentage of Cr-rich rims?

Response: We have now clarified that the increase of percentage of Cr-rich rims correlates with the increase in eruption frequency, magma output rate, explosivity and seismicity at Mt. Etna. Cr-rich zones are found in antecryst rims and phenocryst cores, both in central conduit and eccentric eruptions. We also find Cr-rich zones in antecryst mantles in most eruptions, but we have clarified that we do not include these for statistical calculations, as they might be significantly older than eruption (e.g., ref. 11). We have also stressed that the primitive magma intrusion is a particularly efficient trigger of eccentric eruptions (lines 317-461, 800-890).

Overall, I think that this paper presents some new and interesting data that contribute to our understanding of the magma reservoir at Etna, and more generally to understanding sub-volcanic processes. This paper goes beyond simply presenting some new data in isolation, instead making connections between these data and other information about the volcanic system - in particular, the correspondence between their calculated durations between growth of Cr-rich zones and eruption and the seismic records prior to eruption is intriguing. It is precisely because they are making these connections that I think it's critical to be clear and explicit about which parts of the conclusions are dependent on which assumptions, so that others in the field can accurately incorporate these results into the ongoing discussion in the literature.

Response: Thank you very much!! Please see response to point 3 above.